# The Role of Self-as-Context as a Self-Based Process of Change in Cancer-Related Pain: Insights from a Network Analysis

**DOI:** 10.3390/healthcare13212722

**Published:** 2025-10-28

**Authors:** Evangelia Balta, Flora Koliouli, Lissy Vassiliki Canellopoulos, Vasilis S. Vasiliou

**Affiliations:** 1Department of Psychology, School of Philosophy, National and Kapodistrian University of Athens, 10679 Athens, Greece; evaggeliabalta23@gmail.com (E.B.); lissycanel@gmail.com (L.V.C.); 2School of Early Childhood Education, Aristotle University of Thessaloniki, 54124 Thessaloniki, Greece; fkoliouli@nured.auth.gr; 3School of Psychology, Cardiff University, Cardiff CF10 3AT, UK; 4Department of Psychology, Royal Holloway, University of London, London TW20 0EX, UK

**Keywords:** self-as-context (SAC), psychological flexibility, pain, coping, cancer, processes of change, network analysis

## Abstract

**Background/Objectives**: The dual burden of cancer and pain during chemotherapy can negatively impact individuals’ personal integrity, or the “self”. Yet, coping strategies addressing these dual challenges are rarely employed in cancer-related pain management. Recent findings from evidence-based behavioral models, such as psychological flexibility in pain, highlight the potential role of self-as-context (SAC) as a central coping strategy for adjustment. The aim of this study was to examine the network structure of “conventional” coping strategies, such as active coping, behavioral disengagement, substance use, seeking support, religion, humor, and avoidance (Brief-COPE-8 coping strategies), in relation to “self-based” coping strategies. **Methods**: Individuals diagnosed with cancer, mostly in advanced stages (i.e., II and III), experiencing cancer-related pain (*n* = 135), completed a cross-sectional online study. Participants filled out self-reported questionnaires, including the Brief-COPE, the Psychological Inflexibility in Pain Scale—Greek Version (G-PIPS-II), and the Self-as-Context Scale (SACS) scale, which included two subfactors: centering and transcending. The study employed a stepwise analysis plan. We first conducted a series of traditional correlations, analysis of variance (ANOVA), and hierarchical multiple linear regressions, to examine the predictive role of demographics/clinical characteristics, psychological inflexibility, and SAC (independent variables) on the eight coping strategies (dependent variables). We then selected the highest predictors of coping in cancer-related pain and included them in a network analysis model. In the network analysis, we estimated the LASSO network regularization and examined network stability. We also assessed the centrality and stability of the network model, focusing on the associations between SAC items, the most predictive coping strategies (Brief-COPE), and psychological inflexibility (G-PIPS-II). **Results**: SAC correlated positively with effective coping (active coping and humor) and negatively with substance use. There were no correlations between demographics, type, stage of cancer, and coping strategies for pain. Multiple linear regressions identified psychological inflexibility and SAC as the main contributors to pain adjustment, with SAC explaining substantially more variance in active coping. The partial correlation network included 12 nodes. Active coping, centering, and three of the six transcending items were the most influential in the network. Active coping demonstrated the highest centrality, exerting positive links with SAC items that reflected calm reactions and invariant perspective-taking in response to the pain experience. **Conclusions**: SAC might be considered as a tailored, self-based coping strategy for managing cancer-related pain. Future analog studies should explore the role of integrating self-based perspective-taking strategies to momentarily address cancer-related pain.

## 1. Introduction

More than 55% of individuals with cancer—the disease with the highest mortality worldwide [1,2]—experience pain of moderate to severe intensity [3]. Cancer-related pain is associated with increased daily disability and impaired functioning, including fatigue [4], psychological distress [5,6], and low quality of life [4]. Coping strategies can provide incremental support for individuals with cancer-related pain. Yet, the employment of such coping strategies by individuals or their recommendation by clinicians remains limited [7].

Coping strategies for cancer-related pain include control-based strategies, such as behavioral disengagement [8], substance use [9], and avoidance [9]—all of which exacerbate impairment [10]—and active strategies, such as active coping [11], the use of humor [12], seeking support [13], and religion [14], which enhance functioning. Although a wide range of strategies exist and the literature offers substantial knowledge about coping [15,16,17,18,19], evidence of their efficacy in reducing distress or promoting adjustment to cancer-related pain remains limited [20]. One reason for the modest effects may be conceptual, stemming from the way cancer and pain are studied in relation to “the self”. Historically, studies have examined cancer and pain separately, rather than in the context where pain unfolds [21].

Advancing our understanding of cancer-related coping requires conceptualizing both cancer and pain as contextual stressors that impact individuals’ personal integrity and sense of “self” [22]. In this context, early scholars, such as William James (1981) [23], underscored the importance of discriminating between I (subject) and me (object). From this perspective, the self is not something we are, but something we do—a relational behavior that is developed through the ability to take perspective on one’s own experience [24]. Rather than being a static entity, the self has a contextual and functional role. In this sense, the self can serve a dynamic process of change that supports adaptive functioning, particularly when people face multiple challenges [25]. Thus, self-focused processes (e.g., perspective-taking techniques and skills) may have strong clinical utility.

People with cancer-related pain face multiple challenges, such as side effects of chemotherapy [26], persistent pain [27], and fear of cancer recurrence [28], all of which significantly affect physical and psychosocial well-being. These experiences may prompt identity-related questions (e.g., “the real me”; “Who am I?” [29]), especially during physically and emotionally taxing phases, such as chemotherapy, where pain is prevalent [27]. Cancer pain, in particular, can disrupt one’s sense of identity and provoke uncertainty about who one is or might become [30].

Viewing both cancer and pain as ongoing threats to physical and psychological integrity—or as disruptions to the “threads of the self” [31]—may provide valuable clinical insights into self-based coping strategies that address both experiences simultaneously. Such strategies grounded in the self may help individuals perceive cancer and pain as integral parts of their experience rather than as threats to their identity. Despite this potential, the role of self-based coping in cancer-related pain remains underexplored, partly due to the conceptual complexity of studying the self.

In chronic pain research, multiple terms have been used to describe the self, including self-concept [29], mental defeat [29], self-pain [29], and self-compassion [29]. A recent systematic review identified 15 different self-related constructs, highlighting the potential for confusion in the field [30]. This confusion often arises from the use of different terms for overlapping constructs, each rooted in distinct or overlapping theoretical frameworks [30]. Regardless of terminology, the self is consistently linked to well-being and functioning in chronic pain [30]. Given this conceptual diversity, careful attention should be given to the evaluation of which frameworks most effectively translate into meaningful clinical applications.

Among recent developments, contextual behavioral science provides one such perspective, emphasizing the importance of exploring the sense of self—particularly in relation to chronic pain—within a contextual and functional framework [30]. Contextual behavioral science (CBS) offers a pragmatic approach to understanding complex, interacting experiences, such as cancer and pain [32]. By identifying key processes of change, such as self-as-context (SAC), CBS aims to target mechanisms within specific contexts that can lead to meaningful therapeutic outcomes [33]. Its precision, influence, and scope with regard to the processes of behavioral change make CBS particularly suitable for developing coping procedures with lasting effects [32]. 

From a CBS perspective, the self is conceptualized as a learned behavioral repertoire, examined through the capacity for perspective-taking [34,35]. This aligns with the health model of adaptation, known as psychological flexibility (PF), particularly in the context of chronic pain [34]. PF refers to the ability to adapt thoughts, emotions, and behaviors to changing situations in ways that align with one’s values and goals [36]. PF is moderately associated with well-being and functioning, though research in cancer-related pain is still limited [37]. The model serves the applications of the therapeutic approach of Acceptance and Commitment Therapy (ACT) [38], which offers evidence-informed processes for enhancing psychological flexibility. Within ACT, SAC plays a central role [39]. Studying the self through PF thus may offer clinically meaningful insights and pathways for healthcare professionals to support self-based coping strategies in individuals with chronic pain.

SAC [40] is a core self-based coping process that describes different perspectives of self [41]. It refers to the idea that the self is not limited to thoughts, emotions, and sensations, but reflects a transcendent viewpoint from which individuals can safely observe their ongoing experiences [42]. To illustrate this, when practicing SAC, individuals notice both the experience (e.g., cancer pain) and the person (themselves) who observes it). This separation between “the noticer” and the experience is a trainable skill. Research indicates that SAC is associated with improvements in pain interference [42], daily functioning [43], and mood [44]. Notably, evidence proposes that SAC may be the most influential process within the PF model [45]. This has a clinical implication for cancer-related pain, as multiple stressors can cause ongoing disruptions to the sense of self [46]. To advance research on the self, methods are needed that examine SAC as part of a system of interrelated constructs. Yet its role as a dynamic and central process of change in cancer-related pain remains largely unexplored.

One way to understand the interplay between coping processes and cancer-related pain is through network analysis, which conceptualizes stressors and coping processes as components of a dynamic, mutually influencing system [47]. Unlike conventional statistical models, which often rely on static and linear assumptions, network analysis identifies direct relationships through partial correlations This approach enables the exploration of the most central (i.e., influential) coping responses and dynamic interconnections among variables, called edges [48]. To achieve this, Gaussian Graphical Models (GGMs) are commonly employed. GGMs produce visual representations in which nodes represent variables (e.g., scores from relevant scales or individual items), and edges (lines) illustrate the statistical relationships between them. Edge thickness indicates the strength of the association between two nodes, allowing intuitive interpretation of the network structure and the underlying affective dynamics [49]. These central edges may serve as key targets for clinical interventions.

The aim of this study was to examine the potential contribution of SAC to coping with cancer-related pain. We hypothesized that (a) SAC would be significantly correlated with psychological inflexibility and conventional coping strategies; (b) demographic, such as sex and age, and clinical characteristics, such as the type and the stage of cancer, would be associated with coping strategies and SAC; (c) SAC would positively contribute to conventional coping strategies; and (d) SAC items would exhibit high centrality relative to other coping-related processes, or psychological inflexibility, in the context of cancer and pain.

## 2. Materials and Methods

### 2.1. Design

This study recruited individuals from the community to participate in an online, cross-sectional, descriptive-design study. Before data collection began, participants read and completed an online informed consent form. Due to pandemic-related restrictions, researchers secured digital letters of permission with affixed e-signatures from the participants. Participants were free to refuse to answer any question or withdrawn at any time. Given the sensitive nature of the study, participants were allowed to contact the PI (FK) directly. Data collection took place between October 2021 and February 2022. The study was conducted in accordance with the local legislation, approved by the ethics committee of the Department of Psychology at the National and Kapodistrian University of Athens, Greece (Ref. number updated: 748-11 July 2025), and complied with the Helsinki Declaration of 1975, as revised in 1983.

### 2.2. Study Criteria

Inclusion criteria were as follows: (a) adults, (b) a diagnosis of any type of cancer during the study period, (c) self-report of cancer-related pain. Exclusion criteria, assessed via demographic and clinical questions, were as follows: (a) underage individuals (below 18 years old) with cancer, (b) individuals cured or in remission, and (c) those reporting no cancer-related pain.

### 2.3. Recruitment and Study Procedures

We recruited eligible participants via associations and non-profit organizations in Greece, using convenience and purposive sampling. Recruitment included geographically dispersed associations across both the mainland and islands. For example, among others, we partnered with the Association of Cancer Patients—Volunteers—Friends and Doctors of Athens, the Cyclades Cancer Patients & Friends Association, AgaliaZo—Group of Volunteers Against Cancer of Western Greece, etc., (for the full list of participating organizations, see Appendix A). A total of 135 participants were eligible to participate in this study, of which all the participants completed the questionnaires, and the response rate of the questionnaire was 100%.

### 2.4. Scales

Participants completed a series of scales, including demographics (e.g., age, gender, marital and employment status), clinically relevant characteristics (e.g., cancer onset), and a set of three standardized self-reported questionnaires, measuring conventional and self-based coping strategies.

#### 2.4.1. Conventional Coping Strategies

Coping was measured with the 28-item Brief-COPE [50] (Greek adaptation [51]). This instrument assesses various coping practices across eight sub-factors: active coping, behavioral disengagement, substance use, seeking support, religion, humor, avoidance, and expression of negative feelings. Items are rated on a 4-point scale (1 = not at all to 4 = very much). Subscales scores are calculated separately, as the measure is not intended to produce a single total score. Brief-COPE measures the frequency with which individuals utilize these eight coping strategies in response to difficult situations. Higher scores on a subscale indicate greater frequency of using that particular coping strategy [50]. COPE’s reliability and various aspects of validity, including face, construct, and convergent validity have been reported as satisfactory [51]. Cronbach’s alpha for this study was α’s = 0.85 (total scale), 0.77 (active coping), 0.70 (behavioral disengagement), 0.81 (substance use), 0.75 (seeking support), 0.82 (religion), 0.89 (humor), 0.52 (avoidance), and 0.45 (expression of negative feelings).

#### 2.4.2. Self-Based Coping Strategies

Psychological flexibility: The Greek Version of the Psychological Inflexibility in Pain Scale (G-PIPS-II [52]; English version [53]) is the only scale focusing on the psychological inflexibility in the context of pain and consists of two subscales: avoidance and fusion. Hence, in this study, we used the scale as a general scale of the psychological inflexibility in pain. Items are rated on a 7-point scale (1 = never to 7 = all the time) and a higher total score indicates greater psychological inflexibility. Evidence of reliability and construct, convergent, and discriminant validity has been reported as satisfactory [52]. Cronbach’s alpha for this study was α’s = 0.91, 0.93, and 0.76 for total scale, avoidance, and fusion subscale, respectively.

Self-as-context: SAC scale (SACS) assesses the centering and transcending dimensions of the self [54]. The 10 items are rated on a 7-point scale (1 = completely disagree to 7 = completely agree) and higher scores indicate a greater sense of self-as-context. The centering subscale includes items 1, 2, 5, and 6 and the transcending subscale includes items 3, 4, 7, 9, 10, and 11. No items are reverse-scored. For this study, the SACS was translated and adapted into Greek using WHO forward–back translation criteria and international guidelines for cross-cultural adaptation [55]. Two bilingual researchers conducted the translations, and a third co-author with ACT expertise reviewed the version. The research team resolved discrepancies collaboratively and finalized the Greek version. Psychometric testing indicated a two-factor structure, consistent with the original version: centering (referring to calm reactions to unwanted psychological experiences) and transcending (referring to an invariant perspective-taking of what is characterized “observing self”; [54]). Validity was supported through associations with related questionnaires (e.g., G-CPAQ [56]), and internal consistency was good (α’s = 0.85, 0.73, and 0.78 for total scale, centering, and transcending subscales, respectively).

### 2.5. Power Analysis

The proposed analysis consisted of a maximum of 17 variables, employed for different types of analyses. The first round of analyses included correlations and multiple regressions, and following suggestions from Cohen and Field [57], a G* power 2 analysis [58] for nine predictors and for linear multiple regressions. The findings suggested a total sample size of *n* = 166 participants, with an actual power assumption of 0.95, and an effect size of f^2^ = 0.15, for *p* < 0.05. For the second round of analyses, we expected a maximum of *n* = 15 nodes to occur in the network model. The number of finally collected samples (*n* = 135) allowed us to examine the network accuracy of the resultant network model, including the application of the highly conservative LASSO penalty estimator [59].

### 2.6. Data Analysis

We followed a stepwise approach in the data analyses. We first conducted a series of conventional analyses, using correlations and predictive regression approaches, to identify certain predictors (coping) of managing cancer-related pain. We then used the findings of certain predictors and examined their dynamic links using network analyses. For the analyses, we employed both Jamovi (1.6.23) [60] and then JASP 0.95.2 which integrates the necessary R packages for estimating network structures and assessing model stability [61].

Firstly, the preliminary analysis examined the parametric assumptions, including normality (via histograms and P–P plots) and missing data. Internal consistency was assessed with Cronbach’s α, with 0.70 considered sufficient and 0.80 high [62]. Secondly, Pearson correlations examined the relationships between psychological inflexibility, SAC, and the 11 coping strategies. Thirdly, *t*-tests for independent samples assessed sex differences and One-Way Analysis of Variance (ANOVA) examined if the choice of coping strategies varies by age and clinical variables. Regarding the clinical variables, the ANOVA examined whether the choice of coping strategies varied by type of cancer (e.g., breast, lung, colon cancer, etc.), stage of cancer (e.g., 0, I, II, III, IV), and time of diagnosis (e.g., 1 month to 10 years or more). Then, we conducted a series of hierarchical multiple linear regressions (simultaneous forced-entry method using *R*^2^ and adjusted *R*^2^) to examine the prediction of demographics/clinical characteristics (namely gender, age, marital status, employment, type of cancer, type since diagnosis, stage of cancer) and self-based coping (namely psychological inflexibility and SAC) (nine independent variables) on the eight coping strategies (dependent variables). The findings from the regression analyses defined the variables that we consequently tested in the network analyses.

Following the conventional analyses, we conducted a network analysis. Specifically, we estimated the network structure and node centrality indices using the qgraph package and estimated the network accuracy and stability using the bootnet package.

We first constructed a partial correlation network, in which nodes represent variables and edges represent partial correlation coefficients between pairs of nodes, controlling for all other variables in the network. To estimate this network, we used LASSO regularization (Least Absolute Shrinkage and Selection Operator [63]) in combination with the Extended Bayesian Information Criterion (EBIC) selection method [64], both embedded in JASP [61].

LASSO regularization is recommended for estimating nodes and edges, particularly for partial correlation analyses, as it reduces the likelihood of type I error [65] due to the potential instability of models run in samples with fewer than 500 participants [66]. This approach helps to simplify the results by applying a form of regularization that puts penalties that shrink negligible edge weights to zero, providing more stable and parsimonious solutions for models. Unlike traditional *p*-value-based thresholds, LASSO uses a tuning parameter (lambda) (λ) to remove potentially spurious (false-positive) edges [67].

To further refine model selection, we used EBIC, which includes a hyperparameter (γ). Following conservative practice, we set γ = 0.5 to enhance parsimony by retaining only the most robust edges and shrinking weaker associations to zero [64].

The resultant model was graphically illustrated using the Fruchterman and Reingold algorithm [68], which positions strongly linked nodes about the center and weakly linked nodes at the periphery. For the model, we report only the strength centrality as our main centrality metric, presenting the direct strength of connection (absolute edge weights) among the network’s nodes. Higher centrality defines a more important role (influence) of each node within the model.

We finally used the non-parametric bootstrapped methods, using the bootnet package [69], to assess the network’s accuracy and stability and to detect how consistent the findings were [59]. For the resultant network model, we calculated the 95% coefficient confidence intervals (CIs) for the edge weights and 5000 case-dropping bootstrapped samples to investigate the stability of the indices of centrality.

## 3. Results

### 3.1. Sample, Descriptive Characteristics, and Preliminary Checking

The present sample included *n* = 135 individuals with cancer-related pain, mostly females with a mean age of 46 to 55 years old. Participants reported different types of cancer, such as breast cancer (64.4%, *n* = 87), lung cancer (3.7%, *n* = 5), colon (5.9%, *n* = 8), and other types (26%, *n* = 35), and different stages of cancer such as stage 0 (8.9%, *n* = 12), I (13.3%, *n* = 18), II (22.2%, *n* = 30), III (21.5%, *n* = 29), and IV (5.2%, *n* = 7), while some patients did not wish to answer (11.1%, *n* = 15) and others were in a different status (17.8%, *n* = 24). Participants resided in geographically dispersed areas across the country, including individuals from the north Greece (38%), central Greece (29.8%), and the south (Peloponnese, 23.9%), with a small number residing on the islands (4.4%). We present the participants’ descriptive and main study variable information in Table 1.

No violation of normality was identified in the majority of scales (linearity, homoscedasticity, and collinearity; *W*(135) = 0.97, *p* = 0.002; *W*(135) = 0.99, *p* = 0.548; and *W*(135) = 0.99, *p* = 0.280 for the G-PIPS-II, Brief-COPE, and SAC scale, respectively). Also, we examined the missing values by carrying out a visual inspection and spot checks of the data and this indicated a few cases of random missing data. To confirm our observations, we ran a per item missing value analyses and calculated a Little MCAR test. The per item missing value analysis indicated less than 2% missing data across the variables, which is considered negligible. Also, the MCAR test indicated that data were found to be missing completely at random. Likewise, there were negligible univariate and multivariate outliers; thus, we left the data intact.

### 3.2. Correlation Analyses

Psychological inflexibility was positively correlated (*p* < 0.001) with behavioral disengagement (*r* = 0.31), substance use (*r* = 0.23; *p* < 0.01), avoidance (*r* = 0.36), expression of negative feelings (*r* = 0.35), active coping (*r* = 0.36), seeking support (*r* = 0.20; *p* < 0.05), and religion (*r* = 0.23; *p* < 0.01). There were no correlations between psychological inflexibility and humor (*p* > 0.05). Also, there were no correlations between psychological inflexibility and SAC (*p* > 0.05). Moreover, SAC was (all *p* < 0.001) positively correlated with active coping (*r* = 0.40) and humor (*r* = 0.44) and negatively correlated with substance use (*r* = −0.28). There were no correlations between SAC, behavioral disengagement, seeking support, religion, avoidance, and expression of negative feelings (all *p* > 0.05), as shown in Table 2.

There were no significant relationships between sex, age, and conventional coping strategies. Also, there were no correlations between type and stage of cancer and conventional coping strategies. However, time since diagnosis onset was found to be related to religion as a coping strategy. We found that individuals with a more recent diagnosis, such as a year ago (*M* = 5.65), tended to use religion as a coping strategy more than individuals with a diagnosis nine years ago (*M* = 2.25; *W*(10, 37.3) = 7.37; *p* < 0.001). This indicates that religion as a coping strategy may be a co-variant for individuals with cancer, but this may also be dependable on the context (e.g., the country where the study was conducted). Finally, there were no correlations between self-based coping (SAC) with sex, age, or type and stage of cancer (all *p* > 0.05).

### 3.3. Multivariate Analyses

We conducted eight regression analyses to examine the predictive ability of the demographics/clinical characteristics and the two targeted self-based coping strategies in the eight conventional coping strategies. For all the models examined, the variance inflation factor (VIF) was less than 3.3, and tolerance statistics were all 0.771 or above. As Table 3, below, indicates, for all the eight models we tested, the three predictors accounted for only 29% of the variance explained in active coping strategy (adj. *R*^2^ = 0.29). For this sole statistically significant model, the demographics and clinical characteristics variables were not significant predictors and only psychological inflexibility (*B* = 0.10; *p* < 0.001) and SAC (*B* = 0.18; *p* < 0.001) predicted the use of active coping.

### 3.4. Resultant Network Analysis

#### 3.4.1. Results of Centrality Indices

Figure 1 presents the visual illustration of the resultant network model and Table 4 shows the lasso-regularized values of the edges. Given our primary interest in the central role of self-based coping, we examined each of the ten SACS items individually within the network, focusing on their associations and centrality (i.e., relative influence) in relation to two previously identified predictors: active coping (as measured by the Brief-COPE) and psychological inflexibility (as measured by the G-PIPS-II).

Prior to analyzing the network’s local properties, we visually examined its global structures. As expected, we observed strong interconnections among the SACS items. The most central nodes included items SACS #1, #2, and #6 from the centering sub-factor (strength centrality values = −1.074, 1.222, 1.125, respectively), and #3 and #9 from the transcending sub-factor (strength centrality values = 1.269 and 1.264, respectively). Notably, active coping emerged as the most central node in the network (strength centrality value = −1.564), followed closely by the SACS transcending items #3 (Despite the many changes in my life, there is a basic part of who I am that remains unchanged), and #9 (Even though there have been many changes in my life, I’m aware of a part of me that has witnessed it all), as well as the centering items #6 (I am able to notice my changing thoughts without getting caught up in them) and #2 (I have a perspective on life that allows me to deal with life’s disappointments without getting overwhelmed with them). Psychological flexibility showed medium centrality within the network (strength centrality value = −0.781).

For the resultant model, including the LASSO-regularized partial correlation network, which accounted for the inter-relationships between all nodes, active coping was strongly and positively linked with the centering item #2 (edge value = 0.352) and with the transcending item #9 (edge value = 0.073). Equally, G-PIPS-II was strongly linked with the transcending items #9 and #10 (edge values = 0.174 and 0.238, respectively). As expected, there were multiple partial correlations among the SACS items, particularly between those belonging to the same subfactor (e.g., #3 with #4; #5 with #6, etc.), suggesting a coherent subfactor structure within the network.

Taken together, the findings from the network analysis underscore the importance of maintaining an active coping style that enables individuals to take constructive actions in response to cancer-related pain. Notably, adopting a calm and observant perspective—one that allows individuals to experience the pain without becoming overwhelmed—may support a more mindful and awareness-based perspective-taking of adaptive coping with cancer-related pain.

#### 3.4.2. Network Accuracy and Stability

The Appendix A illustrates the bootstrapped 95% CIs of the edge weights and the stability of centrality indices, respectively. For the edge stability plot (see Appendix A), for most of the edges (connections between variables), we observed shorter lines (CIs) and CIs not including zero. This suggests that each shorted line edge (also not containing zero) is consistently present across the 5000 bootstrap samples. The correlation stability (CS) coefficients (see Appendix A) for most of the edges were above the minimally acceptable index stability (CS > 0.25), suggesting high index stability. However, when we visually inspected the boxplots for the strength’s stability, we observed some steady drop distribution of correlations between the original centrality values and the values obtained from case-dropping bootstrap samples (the correlation should be retained at least 0.7 as cases are dropped). Specifically, the Spearman’s rho median correlation (e.g., the y-axis representing the difference between original and bootstrapped centrality value correlations) was 0.41 for active coping, 0.62 for the psychological inflexibility (G-PIPS-II), and for the key SACS items 2, 9, and 10 were 1.10, 1.20, and 0.71, respectively. These findings indicate some instability in the centrality matrix, proposing a cautious interpretation of the resultant network models. The mean predictability of the overall network was 0.825.

## 4. Discussion

The aim of this study was to examine the contribution of SAC in the context of cancer-related pain. Specifically, we investigated the central role of SAC as a self-based coping strategy among individuals with cancer-related pain. For this proof-of-concept study, we employed a stepwise analytic approach. We first identified predictors of coping and then constructed a network model of associations. The study presents significant findings about the relationship between coping strategies for cancer-related pain and adjustment that can have multiple benefits for the psychosocial support of individuals battling cancer, including tailoring the components of interventions for managing cancer-related pain, optimizing existing interventions, and enhancing personalization in cancer-related pain. To our knowledge, this is the first study to investigate the nuanced role of SAC in this context.

Firstly, despite theoretical and empirical links between psychological flexibility, SAC, and improved adjustment to pain [42], we observed no association between SAC and psychological inflexibility. This does not necessarily undermine the coherence of the model [70,71,72]. Rather, it indicates the effect of the selection of the G-PIPS-II scale, which captures more aspects of avoidance and fusion than the whole six processes of psychological inflexibility [52]. Our findings are consistent with previous studies [10,36,73], indicating that individuals with low levels of psychological flexibility use avoidance coping strategies, but also positive coping strategies. Thus, what is often labeled as “inflexible” coping may, in certain contexts, serve functional and adaptive purposes, combining both positive and negative aspects. Contrary to previous research, sociodemographic and cancer-related variables—such as sex, age, cancer stage or type—were not significant predictors of coping strategy use in this study [74]. Individuals with a more recent cancer diagnosis were more likely to use religion as an active coping strategy; this could possibly reflect cultural factors, such as the centrality of religiosity as a coping mechanism [73], but may also point to preferences in coping approaches [75]. Lastly, contrary to previous research indicating an association between disease type and stage and certain coping strategies used [76], this was not observed in the present sample. The minimal predictive value of demographic and clinical characteristics highlights the importance of focusing instead on psychological processes—such as SAC—in understanding coping and adjustment to cancer pain. Our regression and network analyses emphasized the potential clinical relevance of SAC as a self-coping process of change in coping with cancer-related pain. Specifically, our findings indicate that SAC may offer some distinct advantages over other adaptive coping mechanisms, such as humor and active coping—a concept that has received limited attention in the literature to date [73]. This observation arises because of the centrality some of the SAC items presented and is consistent with earlier studies that have identified SAC’s unique contribution to emotional functioning [73]. 

From the network perspective, among the 66 edges (connections among all variables), more than half of them (37/66) presented with a connection in the LASSO network, with most of the SACS items exerting a central role. Despite the modest sample size and the fair observed stability of the network, we can make some interpretations about the centrality of certain nodes and their edges that still need to be considered with caution.

Firstly, we found that active coping and the centering sub-factor both presented strong positive conditional associations with two distinct items of the SACS: the centering item #2 (I have a perspective on life that allows me to deal with life’s disappointments without getting overwhelmed with them) and the transcending item #9 (Even though there have been many changes in my life, I’m aware of a part of me that has witnessed it all). Additionally, we found a negative conditional association between psychological inflexibility and the transcending items #9 and #10 (Even though there have been many changes in my life, I’m aware of a part of me that has witnessed it all and I am able to access a perspective from which I can notice my thoughts, feelings, and emotions). The findings are in alignment with a new line of network analysis research that positions SAC as a central (therefore more influential) process that extends beyond psychological flexibility alone and further explanation is provided below [77].

Our findings suggest that SAC may provide incremental support in coping with cancer-related pain by prompting individuals to engage in more conventional coping strategies, particularly active coping. The mechanism underlying this contribution may involve individuals attempting to contextualize their pain within the broader context of their cancer experience, separating the experience itself (e.g., “this excruciating pain is just a part of my cancer experience- does not define me”) from its defining impact (“this excruciating pain is killing me”). Indeed, our findings indicated that the strongest weight edges (partial LASSO network links) were found between active coping and two specific SACS items reflecting self-based behaviors of calm reactions to unwanted experiences (SACS 2; I have a perspective on life that allows me to deal with life’s disappointments without getting overwhelmed with them) and an ongoing awareness of the self in the context of an unwanted experience (SACS 9; Even though there have been many changes in my life, I’m aware of a part of me that has witnessed it all). This invariant perspective-taking may help individuals decenter from pain and redirect their energy toward more adaptive actions.

Likewise, the observed negative association between psychological inflexibility and items from the transcending sub-scale (#9 and #10) may reflect theoretically grounded mechanisms, involving self-based processes of change. Specifically, this finding may indicate the contribution of healthy “selfing” practices, including self-as-process (i.e., an ongoing discrimination of our behaviors as they occur in the moment [25]), and self-as-context (i.e., adopting a transcendent perspective of the self as a stable observer of internal experiences [25]) in the management of cancer-related pain. From this perspective, SAC may indirectly foster adaptive coping—particularly active coping—by enabling individuals to contextualize, i.e., recognize (verbally discriminate [25]) and label, their ongoing experiences more flexibly (e.g., “right now I am experiencing cancer-related pain that feels debilitating”) and within the broader cancer experience. For example, reframing pain as “a part of the cancer experience, but not defining the self” may facilitate decentering from pain and redirecting energy toward constructive action. Similarly, transcendent awareness may buffer against rigid, inflexible responses by enabling individuals to notice and label experiences without being consumed by them. Such self-based processes may help preserve a stable sense of identity during times when cancer and pain threaten the “threads of the self” [28]. There is a need for more experiential and analogous proof-of-concept studies to support this.

Notwithstanding its strengths, this study has its limitations, such as a low representation of men, small sample size, great heterogeneity of clinical characteristics, and the self-report survey, which all partly reflect response and common method biases. Furthermore, the cross-sectional design indicates that we should not draw any causal inferences about the role of SAC in alleviating cancer-related distress due to pain. Additionally, the convenience-sampling approach, recruiting individuals with cancer from national charities and supporting groups, limits the variability of individuals with cancer pain. Moreover, while cultural influences—such as religiosity—were discussed as potential explanatory factors, we did not directly measure cultural or religious variables. This decision was based on the relatively culturally homogeneous nature of the Greek population, but it limits the ability to generalize findings to more multicultural or diverse settings, where cultural frameworks may differently shape coping strategies.

Despite the aforementioned limitations, this study facilitates a better understanding of the role of SAC in cancer-related pain management. Further research should now particularly examine how personalized, self-based, and culturally sensitive coping responses can provide a tailored support to an individual’s needs. This requires advanced methods, such as an Ecological Momentary Assessment (EMA) or idiographic type of analyses, that look at the intraindividual level; hence, they can more precisely indicate how individuals can apply self-based coping responses in specific contexts. Also, future research should now test the development of foci/personalized interventions for individuals experiencing cancer pain.

### Implications for Psychosocial Providers or Policy

Our findings underscore that certain individuals with cancer pain may benefit from more personalized approaches, perhaps as part of a stepped care model [78]. Considering the high heterogeneity of individuals with cancer, innovative approaches like Process-Based Therapy (PBT) [79] should be the next focus of psychosocial interventions for cancer pain. PBT involves personalizing treatment components to address targeted biopsychosocial processes relevant to individual treatment goals. Embracing such idiographic and personalized psychosocial approaches lays the groundwork for clinically pragmatic research in personalized psychosocial management for cancer pain. Such advancements, heralded as breakthroughs in cancer pain management (BtCP) [80,81], hold great promise for both clinicians and patients alike.

## 5. Conclusions

This proof-of-concept study provides preliminary evidence for the role of self-as-context in coping with cancer-related pain. SAC appears to complement conventional coping strategies by fostering more flexible, engaged, and adaptive responses, particularly active coping. Encouraging patients to cultivate a stable and adaptive sense of self in the face of distress may buffer against the identity disruptions caused by cancer and its treatment. While replication and extension in larger, more diverse samples are needed, these findings suggest that SAC holds promise as a novel self-based coping process in cancer pain management.

Exploring how cancer and pain can disrupt self-identity may contribute to the development of self-based coping strategies, which could eventually be considered as part of a comprehensive approach to pain management in cancer care. 

## Figures and Tables

**Figure 1 healthcare-13-02722-f001:**
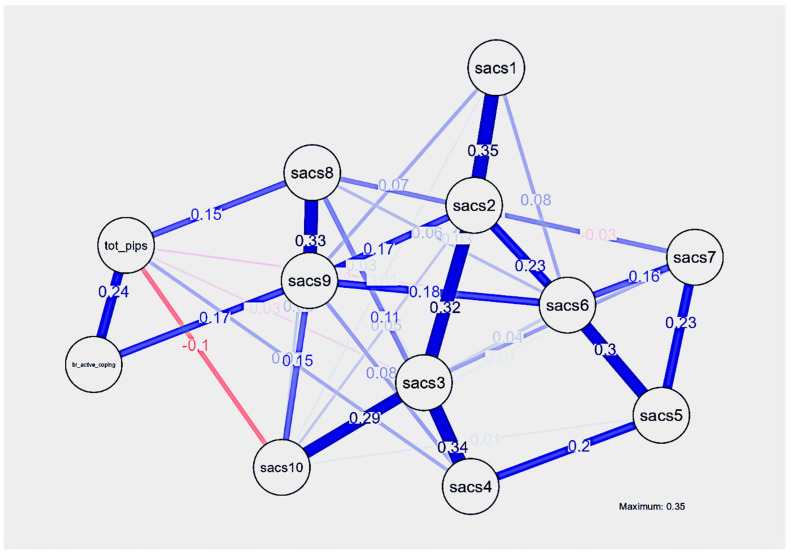
The resultant network analysis showing the LASSO-regularized partial coefficients between SACS items with G-PIPS total scores and COPE active coping scores. Notes: Edge thickness represents the strength of connection between two nodes, with a thicker line representing a stronger connection. Edge color represents the valence of the relationship; blue represents a positive relation while red represents a negative relation. The LASSO regularization does not allow direct interpretation of the physical distance between the nodes (it is only for the visual representation of the nodes). Likewise, the physical spatial information among the nodes and edges within the model does not correspond to quantitatively useful information.

**Table 1 healthcare-13-02722-t001:** Participant descriptive information and study variables (*n* = 135).

Descriptive and Study Variables	Μ ± SD or *n*	Score Range or %
Gender		
Male	11	8.1
Female	124	91.9
Age		
26 to 35 years old	8	5.9
36 to 45 years old	32	23.7
46 to 55 years old	59	43.7
56 to 65 years old	25	18.5
66 years old and older	11	8.2
Marital status		
Single	8	5.9
In relationship	7	5.2
Married or cohabiting without children	8	5.9
Married or cohabiting with children	83	61.5
Divorced	21	15.6
Widow/widower	7	5.2
Other status	1	0.7
Employment status		
Employee	69	51.1
Unemployed	16	11.9
University student	2	1.5
Retired	25	18.5
Household	21	15.6
Do not wish to answer	1	0.7
Other status	1	0.7
Time since diagnosis		
Within the last 5 years	84	62.6
6 to 9 years	29	21.7
10 years and more	21	15.7
Brief—COPE	66.5 ± 12.3	28–101
Active coping	23.3 ± 4.84	8–32
Behavioral disengagement	2.66 ± 1.30	2–8
Substance use	2.24 ± 0.86	2–8
Seeking support	9.83 ± 3.19	4–16
Religion	4.97 ± 2.18	2–8
Humor	5.64 ± 2.13	2–8
Avoidance	8.69 ± 2.60	4–16
Expression of negative feelings	9.15 ± 2.57	4–14
Psychological Inflexibility in Pain Scale	48.2 ± 17.3	17–84
Self-as-context scale	49.4 ± 10.8	16–70

**Table 2 healthcare-13-02722-t002:** Correlations between main study variables.

	1.	2.	3.	4.	5.	6.	7.	8.	9.	10.
1. G-PIPS-II	-									
2. SACS	0.04	-								
3. Active coping subscale	0.36 ***	0.40 ***	-							
4. Behavioral disengagement subscale	0.31 ***	0.03	−0.04	-						
5. Substance use subscale	0.23 **	−0.28 ***	−0.14	0.55 ***	-					
6. Seeking support subscale	0.20 *	−0.04	0.43 ***	−0.04	−0.03	-				
7. Religion subscale	0.23 **	0.08	0.30 ***	0.03	0.00	0.29 ***	-			
8. Humor subscale	0.01	0.44 ***	0.43 ***	0.01	−0.09	0.19 *	0.17 *	-		
9. Avoidance subscale	0.36 ***	0.15	0.51 ***	0.29 ***	0.14	0.30 ***	0.10	0.25 **	-	
10. Expression of negative feelings Subscale	0.35 ***	0.11	0.43 ***	0.22 *	0.15	0.44 ***	0.30 ***	0.39 ***	0.35 ***	-

* *p* < 0.05, ** *p* < 0.01, *** *p* < 0.001.

**Table 3 healthcare-13-02722-t003:** Hierarchical linear regression models.

	Standardized Beta						
Variables	Β	t	P	R2	Adj. R^2^	F Change	Sig. F Change (*p*)
Dependent variable: COPE: Active coping							
Gender	−0.20	−0.13	0.90				
Age	−0.45	−1.12	0.26				
Marital status	0.17	0.47	0.64				
Employment status	−0.31	−1.24	0.22				
Type of cancer	−0.15	−1.60	0.11				
Time since diagnosis	−0.01	−0.12	0.91				
Stage of cancer	−0.05	−0.26	0.80				
Psychological inflexibility in pain scale	0.10	4.67	<0.001	0.29	0.25	7.35	**<0.001**
Self-as-context scale	0.18	5.12	<0.001	0.29	0.25	7.35	**<0.001**
Dependent variable: COPE: Behavioral disengagement							
Gender	0.36	0.84	0.40				
Age	−0.03	−0.22	0.83				
Marital status	−0.03	−0.28	0.78				
Employment status	0.12	0.07	0.10				
Type of cancer	0.03	0.99	0.32				
Time since diagnosis	−0.03	−0.76	0.45				
Stage of cancer	0.01	0.10	0.92				
Psychological inflexibility in pain scale	0.02	3.45	<0.001	0.12	0.06	1.91	0.06
Self-as-context scale	0.00	0.42	0.68	0.12	0.06	1.91	0.06
Dependent variable: COPE: Substance use							
Gender	0.22	0.76	0.45				
Age	−0.04	−0.46	0.65				
Marital status	−0.04	−0.58	0.57				
Employment status	0.07	1.57	0.12				
Type of cancer	−0.00	−0.27	0.79				
Time since diagnosis	−0.01	−0.46	0.65				
Stage of cancer	−0.00	−0.12	0.91				
Psychological inflexibility in pain scale	0.01	0.00	0.008	0.10	0.04	1.68	0.11
Self-as-context scale	−0.02	0.01	0.003	0.16	0.10	2.61	**0** **.01**
Dependent variable: COPE: Seeking support							
Gender	−0.23	−0.20	0.84				
Age	−0.19	−0.55	0.58				
Marital status	−0.23	−0.87	0.38				
Employment status	−0.03	−0.16	0.88				
Type of cancer	−0.03	0.41	0.69				
Time since diagnosis	−0.23	−2.81	0.006				
Stage of cancer	0.06	0.37	0.71				
Psychological inflexibility in pain scale	0.03	2.12	0.04	0.06	2.14	0.04	0.06
Self-as-context scale	−0.01	−0.36	0.72	0.06	1.91	0.06	0.06
Dependent variable: COPE: Religion							
Gender	0.54	−0.70	0.49				
Age	−0.15	−0.61	0.54				
Marital status	−0.22	−1.22	0.22				
Employment status	−0.08	−0.66	0.51				
Type of cancer	0.06	1.23	0.22				
Time since diagnosis	−0.08	−1.52	0.13				
Stage of cancer	−0.02	−0.15	0.88				
Psychological inflexibility in pain scale	0.03	2.40	0.02	0.11	0.06	1.98	0.05
Self-as-context scale	0.02	0.95	0.35	0.12	0.06	1.86	0.06
Dependent variable: COPE: Humor							
Gender	−0.73	−1.04	0.30				
Age	−0.33	−1.53	0.13				
Marital status	0.05	0.30	0.77				
Employment status	−0.001	−0.01	0.99				
Type of cancer	−0.11	−2.51	0.013				
Time since diagnosis	−0.002	−0.04	0.97				
Stage of cancer	0.05	0.57	0.57				
Psychological inflexibility in pain scale	0.002	0.19	0.85	0.03	−0.03	0.49	0.87
Self-as-context scale	0.09	5.81	<0.001	0.24	0.18	4.29	**<** **0** **.001**
Dependent variable: COPE: Self-distraction							
Gender	0.21	0.24	0.81				
Age	0.02	0.07	0.94				
Marital status	−0.09	−0.43	0.67				
Employment status	0.02	0.14	0.89				
Type of cancer	0.05	0.96	0.34				
Time since diagnosis	−0.007	−0.11	0.91				
Stage of cancer	−0.09	−0.77	0.44				
Psychological inflexibility in pain scale	0.05	4.01	<0.001	0.14	0.09	2.62	**0** **.01**
Self-as-context scale	0.03	1.53	0.13	0.16	0.10	2.62	**0** **.01**
Dependent variable: COPE: Self-blame							
Gender	1.21	1.39	0.17				
Age	−0.32	−1.20	0.23				
Marital status	0.004	0.02	0.98				
Employment status	−0.04	−0.29	0.77				
Type of cancer	−0.03	−0.59	0.56				
Time since diagnosis	−0.04	−0.61	0.55				
Stage of cancer	0.04	0.38	0.71				
Psychological inflexibility in pain scale	0.05	4.23	<0.001	0.17	0.12	3.20	**0** **.002**
Self-as-context scale	0.03	1.33	0.19	0.18	0.12	3.06	**0** **.002**

Notes: We present only the values for the variables of interest to avoid confusion. Statistically significant *p* values (*p* < 0.05) are presented in bold.

**Table 4 healthcare-13-02722-t004:** Resultant LASSO Regularized Partial Correlation Coefficients for all variables examined.

Network Weights												
Variable	Total G-PIPS-II	BRIE (Active Coping)	SACS1	SACS2	SACS3	SACS4	SACS5	SACS6	SACS7	SACS8	SACS9	SACS10
Total G-PIPS-II	0.000	0.000	0.000	0.000	0.000	0.000	0.000	0.000	0.000	0.000	**0.174**	**0.238**
BRIE (Active Coping)	0.000	0.000	0.012	**0.352**	0.029	0.000	0.000	0.076	0.000	0.000	0.073	0.000
SACS1	0.000	0.012	0.000	0.045	**0.288**	0.000	0.013	0.044	0.006	0.054	**0.145**	**−0.103**
SACS2	0.000	0.352	0.045	0.000	**0.325**	0.000	0.000	**0.225**	−0.034	0.000	**0.167**	0.000
SACS3	0.000	0.029	0.288	0.325	0.000	**0.340**	0.000	0.043	0.085	0.025	0.000	−0.026
SACS4	0.000	0.000	0.000	0.000	0.340	0.000	0.202	0.000	0.000	**0.111**	0.084	0.074
SACS5	0.000	0.000	0.013	0.000	0.000	0.202	0.000	**0.302**	**0.232**	0.000	0.000	0.000
SACS6	0.000	0.076	0.044	0.225	0.043	0.000	0.302	0.000	0.162	0.058	**0.182**	−0.030
SACS7	0.000	0.000	0.006	−0.034	0.085	0.000	0.232	0.162	0.000	**0.110**	0.000	0.000
SACS8	0.000	0.000	0.054	0.000	0.025	0.111	0.000	0.058	0.110	0.000	**0.334**	**0.148**
SACS9	0.174	0.073	0.145	0.167	0.000	0.084	0.000	0.182	0.000	0.334	0.000	0.000
SACS10	0.238	0.000	−0.103	0.000	−0.026	0.074	0.000	−0.030	0.000	0.148	0.000	0.000

## Data Availability

The data that support the findings of this study are available from the corresponding author, upon reasonable request.

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
