# Peer review of "The Role of Self-as-Context as a Self-Based Process of Change in Cancer-Related Pain: Insights from a Network Analysis"

_healthcare, 2025, doi:10.3390/healthcare13212722_

Round 1
Reviewer 1 Report
Comments and Suggestions for Authors
Dear authors:
I have reviewed your manuscript entitled “The Role of Self-as-Context as a Self-based Coping with Cancer-Related Pain: Insights from a Network Analysis.” This cross-sectional online study aims to explore the network structure of both “conventional” coping strategies—such as active coping, behavioural disengagement, substance use, seeking support, religion, humour, and avoidance (as assessed via the Brief-COPE, focusing on 8 coping strategies)—and “self-based” coping strategies, including Self-as-Context (SAC) (measured using the SACS scale with its two subcomponents: centring and transcending), along with the role of psychological flexibility (G-PIPS-II).
First, I would like to commend you on choosing such a relevant and meaningful topic. The paper contributes valuable insights to the field of cancer and pain management.
As an external reviewer, I would like to offer the following suggestions for your consideration:
- Abstract: Please ensure the type of study, sample details, and data collection instruments are clearly described in the Methods section of the abstract.
- Methods: Could you kindly clarify the following:
- a) Please clarify the estimated number of individuals in the target population (i.e., members of cancer-related associations and non-profit organizations who could potentially participate in your study). Additionally, specify which associations and organizations you are referring to. This is important, as coping strategies may vary significantly depending on the type of cancer— for example, individuals with head and neck cancer may experience and manage pain differently compared to those with breast cancer. Such variations could influence coping mechanisms and potentially affect the interpretation of your results.
- b) The geographical and cultural context of the community involved, to provide a better understanding of any cultural or religious influences on coping strategies.
- c) Kindly indicate the Principal Investigator (PI) in the appropriate section (line 165, page 4).
At present, I have no further comments. I sincerely wish you the best with the publication of your manuscript, which I found both informative and engaging.
Best Regards.
Author Response
Assistant Prof. Vasiliki Eftimiou,
Special Editor-in-chief HealthCare
London, 28/07/2025
Dear Ass. Prof. Eftathiou,
RE: Re-Submission of a revised version of the Manuscript 3700148
We would like to thank you for inviting us to revise and resubmit our work. We have addressed all of your comments, and we do hope that this revision would allow you to reach a final decision. We would also like to thank the reviewers for their time spent in reviewing this manuscript and their comments which have much improved our manuscript.
Please find below a detailed response to all the reviewers’ comments. We present any changes in the manuscript with a red color font.
Kind regards,
Vasilis S. Vasiliou, Ph.D. (on behalf of all authors)
CPsychol (BPS), Registered Clinical & Health Psychologist (HCPC)
Lecturer in Clinical Psychology
|
Reviewer 1 |
|
|
1. Please ensure the type of study, sample details, and data collection instruments are clearly described in the Methods section of the abstract. |
Thanks for this comment. We agree with this comment and so we added the following in the abstract (page 1, lines 25-29)“ Individuals diagnosed with cancer, mostly in advanced stages II and III, experiencing cancer pain, (n = 135) completed a cross-sectional online study, filling out self-reported questionnaires, including the Brief-COPE, the Psychological Inflexibility in Pain Scale – Greek Version and the Self-as-Context Scale”. |
|
2. Please clarify the estimated number of individuals in the target population (i.e., members of cancer-related associations and non-profit organizations who could potentially participate in your study). |
Thanks for this comment, we did not request a certain quota from each partnering association, but we asked them to help us with the dissemination of the study. |
|
3. Specify which associations and organizations you are referring to. |
We specified the associations and organizations in page 5 – line 194-199 and in the supplementary material 5. Specifically, we recruited participants via various associations being geographically dispersed both in the mainland and the islands of Greek territory. For example, among others, we partnered with the Association of Cancer Patients – Volunteers- friends and Doctors of Athens, the Cyclades Cancer Patients & Friends Association, AgaliaZo- Group of Volunteers Against Cancer of Western Greece, etc (for the full list of participating organization see Supplementary material S4). |
|
4. The geographical and cultural context of the community involved, to provide a better understanding of any cultural or religious influences on coping strategies. |
Thank you. We have accordingly added this text (page 7, lines 324-327): “Participants resided in geographically dispersed areas across the country, including individuals from the North Greece (38%), the Central Greece (29,8%), in the South (Peloponnese, 23,9%), and a small part residing in the islands (4,4%). We present the participants’ descriptive and main study variable information in Table 1. |
|
5. Kindly indicate the Principal Investigator (PI) in the appropriate section (line 165, page 4). |
This was added on page 4, line 179. “…participants were allowed to contact the PI (FK) of this research study.” |

Reviewer 2 Report
Comments and Suggestions for Authors
Brief Summary
This cross-sectional study examines the relationship between self-as-context, conventional coping strategies, and psychological flexibility. A total of 135 adults with cancer and cancer-related pain participated in the study, completing a range of self-report questionnaires. The paper uses a stepwise analysis approach to determine which items to include in the network analysis. It provides a strong rationale for focusing on the sense of self and psychological flexibility in relation to cancer-related pain, and the research appears timely.
General comments
- This study includes a diverse sample with a high completion rate.
- The language adaptation of the scale is impressive.
- Although the main aim centres on the network analysis, it could be highlighted more clearly how the initial analyses informed this approach, to better link the two results sections.
- The results of the network analysis could be presented more clearly, as there were several errors in the figures.
- The discussion could be expanded, particularly regarding its main conclusion suggesting that self-as-context may complement conventional coping strategies.
Specific comments
- Ensure consistent use of the term “substance use” throughout the manuscript. For example, on page 2, line 59, it is referred to as “substance misuse.”
- Consider including a definition of “Self” at the beginning of the introduction. Currently, a definition is introduced as part of the concept of CBS (line 104), but an earlier introduction may help readers.
- Please check all in-text references to ensure that they correspond to the correct papers. I noted that most references do not align with the cited paper in the text. I suspect that some references were added at a later date without updating the bibliography accordingly. For example, references to concepts such as mental defeat or self-compassion cite reference 26, but these concepts are not mentioned in that article and do not cite the original research introducing these concepts.
- Similarly, on page 2, line 87, there is mention of a systematic review identifying 15 self-related constructs, but reference 27 does not appear to cover this. In the last paragraph on page 2, multiple citations refer to reference 27, which does not seem to cover the topics discussed.
- Please also check that all references are complete (e.g., reference 32) and that none are underlined (e.g., references 44 and 48).
- In Hypothesis (a), it is stated that SAC would exhibit a significant correlation with psychological inflexibility and conventional coping strategies. It would be helpful to clarify in the text why this is hypothesised, and the expected direction of the correlation, given the limited research in this area. Similarly, Hypothesis (b) is unclear regarding the direction of the association – for example, what demographic or clinical characteristics are referred to, and does this pertain to general use of coping strategies or the ability to self-cope?
- Regarding psychological flexibility: in the hypotheses and text, this is referred to as “psychological inflexibility.” Please clarify whether this refers to a subset of the original scale or the overall scoring.
Methods
- Page 5, line 188. For the Brief COPE, please provide more information on scoring. What does the summed score indicate? Does a higher score represent greater coping ability? Is there an overall score or are only subscale scores used?
- Power analysis (page 5): Was the sample size calculation based on correlations or multiple regression? Please include additional details such as effect size and statistical power assumptions.
- What approach was used to handle missing data? (Line 243)
- Line 249: Please specify which clinical variables were assessed, with examples.
Results
- For the sample description, please report mean age and standard deviation rather than only the age range (line 291).
- Significance values are reported as “ps,” but they are typically denoted as “p.”
- Section 3.3: It seems odd to include multivariate analyses of other COPE factors if they were not considered variables of interest (line 337). Please clarify the rationale.
- Figure 1: Could you include a legend within the figure to indicate what the different labels refer to, so readers do not need to refer back to the text?
- Supplementary Figures (S3 and S4): These are difficult to interpret. In S3, the legend does not indicate which nodes are being referred to, and in Figure on page 19, the numbers are not fully visible. Please improve figure clarity and include legends.
- Centrality measures table: Are these values standardised and sorted (e.g., from least to most expected influence)? To further improve clarity, could you include a line or indicator connecting the different data points?
Discussion:
- In the discussion, it is stated that both active coping and three out of six transcending items are identified as the most central nodes (line 446), whereas in the results, active coping is described as the “most central node,” followed by four transcending items. Please ensure consistency between results and discussion.
- It would be beneficial to include further discussion on the association between psychological inflexibility and transcending items #9 and #10 (line 451), as this is a key part of the study’s conclusions.
Author Response
|
Reviewer 2 |
|
|
1. Ensure consistent use of the term “substance use” throughout the manuscript. |
Thank you for pointing this out. We have corrected our text using the term “substance use”. (page 2, line 62) |
|
2. Consider including a definition of “Self” at the beginning of the introduction. Currently, a definition is introduced as part of the concept of CBS (line 104), but an earlier introduction may help readers. |
Thanks for this. There are multiple definitions for the self in the literature and for this study we selected to include the following one on page 2, lines 74-80: In this context, early scholars, such as William James (1981) underscored the important of discriminating between I (subject) and me (object). From this stance, the self is not something we are, but something we do- a relational behavior that is developed early one from an individual’s ability to take perspective on its own experience (Hayes et al., 2001). Rathen than being a static entity, the self is a contextual and functional process that can support adaptive functioning, particularly when people face multiple challenges (Mchugh et al., 2019).
Please note that after requesting suggestions for the definition of the self from scholars (Prof. Steven Hayes, personal communication, 19th of July, 2025) we decided to alter the word “coping” with the word “process” in the title as this allies more with both the definition of the self and the type of analyses (network) we followed.
Ref: James, W. (1981). The principles of psychology. Cambridge, MA: Harvard University Press. Hayes, S. C., Barnes-Holmes, D., & Roche, B. (2001). Relational Frame Theory: A Post-Skinnerian Account of Human Language and Cognition. Springer. McHugh, L., Stewart, I., & Almada, P. (2019). A contextual behavioral guide to the self: Theory and practice. New Harbinger Publications. |
|
3. Please check all in-text references to ensure that they correspond to the correct papers. |
Thank you for highlighting it. We have revised our references and they correspond to the correct papers. |
|
4. References to concepts such as mental defeat or self-compassion cite reference 26, but these concepts are not mentioned in that article. |
We agree with this comment, and we changed references to correspond with the correct resources (page 3, line 97, reference 29). |
|
5. On page 2, line 87, there is mention of a systematic review identifying 15 self-related constructs, but reference 27 does not appear to cover this |
We agree with this comment, and we changed references to correspond with the correct resource (page 3, line 99, reference 30). |
|
6. Check that all references are complete (e.g., reference 32) |
We have reviewed our references and now all of them are complete. Thank you for your highlighting it. |
|
7. Check that none are underlined (e.g., references 44 and 48). |
Thank you for pointing this out. We have corrected all our references. |
|
8. In Hypothesis (a), it is stated that SAC would exhibit a significant correlation with psychological inflexibility and conventional coping strategies. It would be helpful to clarify in the text why this is hypothesised, and the expected direction of the correlation, given the limited research in this area. |
We agree that there is limited research examining the associations between SAC and psychological flexibility in the context of cancer pain, yet there is available literature examining these associations in other conditions, particularly chronic pain*.
With this in mind, our first hypothesis examined theory and data driven relationships, expecting negative correlations between coping strategies and psychological inflexibility/ SAC. The hypothesis was partially confirmed with the correlation between SAC and PIPS being non-significant. We provide a thorough explanation in the discussion about this lack of significant finding which also corresponds with other scholars’ findings (page 15, lines: 453-463).
* a list of references below: - Davey, A., Chilcot, J., Driscoll, E., & McCracken, L. M. (2020). Psychological flexibility, self-compassion and daily functioning in chronic pain. Journal of contextual behavioral science, 17, 79-85. - Kwok, S. S. W., Chan, E. C. C., Chen, P. P., & Lo, B. C. Y. (2016). The “self” in pain: the role of psychological inflexibility in chronic pain adjustment. Journal of behavioral medicine, 39(5), 908-915. - Yu, L., Norton, S., & McCracken, L. M. (2017). Change in “self-as-context”(“perspective-taking”) occurs in acceptance and commitment therapy for people with chronic pain and is associated with improved functioning. The Journal of Pain, 18(6), 664-672. - Sirois, F. M., Molnar, D. S., & Hirsch, J. K. (2015). Self-compassion, stress, and coping in the context of chronic illness. Self and Identity, 14(3), 334-347. |
|
9. Hypothesis (b) is unclear regarding the direction of the association – for example, what demographic or clinical characteristics are referred to, and does this pertain to general use of coping strategies or the ability to self-cope? |
Thank you for this comment. As for the first part of the question- the directions of the association, we have now amended this hypothesis to make it more specific. On page 4, lines 164-165 we added the exact variables we examined: “b) demographic, such as sex and age, and clinical characteristics, such as the type and the stage of cancer”. As for the second part of the question-, we report the findings for the associations of demographics/ clinical characteristics with conventional coping and with SAC (the self-coping (SAC). We have now added the term “conventional” coping in line 353 (page 10) and also the results from the correlations between self-based coping (SAC) with the demographics (sex and age) and the clinical characteristics (type and stage of cancer) on page 10, lines 360-362. |
|
10. Regarding psychological flexibility: in the hypotheses and text, this is referred to as “psychological inflexibility.” Please clarify whether this refers to a subset of the original scale or the overall scoring. |
We refer to the overall sum scoring of the 12 items of the G-PIPS, assessing psychological inflexibility. |
|
11. Page 5, line 188. For the Brief COPE, please provide more information on scoring. |
We added more information about the scoring of Brief-COPE (page 5 – lines 213-217). Specifically: “Brief-COPE measures the frequency with which individuals utilize these eight coping strategies in response to difficult situations. Higher scores on a subscale indicate greater frequency of using that particular coping strategy. So individual sum scores for each of the subscales are utilized instead of an overall total score for the entire questionnaire.” |
|
12. Power analysis (page 5): Was the sample size calculation based on correlations or multiple regression? Please include additional details such as effect size and statistical power assumptions. |
As we indicate on page 6 – lines 255 -256, 2.5. Power Analysis, a G* power analysis for nine predictors and for linear multiple regressions. Findings suggested a total sample size of n= 166 participants, with actual power assumption 0.95, effect size f2= 0.15, for p. < .05. |
|
13. What approach was used to handle missing data? (Line 243) |
Visual inspection and spot check indicated a few cases with random missing data. To confirm our observations, we run a per item missing value analyses and calculated a Little MCAR test. The per item missing value analysis indicated less than 2% of missing across the variables which is considered negligible. Also, the MCAR test indicated that data were found to be missing completely at random. We report this finding on pages 7-8, lines 330-336. |
|
14. Line 249: Please specify which clinical variables were assessed, with examples. |
On page 6, line 274-276 we report the following: “Regarding the clinical variables ANOVA examined if the choice of coping strategies varies by type of cancer (e.g., breast, lung, colon cancer etc), stage of cancer (e.g., 0, I, II, III, IV) and time of diagnosis (e.g., 1 month to 10 years and more).” |
|
15. For the sample description, please report mean age and standard deviation rather than only the age range (line 291). |
The question about the age of the participants was not open-ended, but there were these choices: 18 to 25, 26 to 35, 46 to 55, 56 to 65, 66 years old and older. So there was a mean age of 46 to 55 years old that is referred to page 7 - line 319 . |
|
16. Significance values are reported as “ps,” but they are typically denoted as “p.” |
Thank you for pointing this out. We have corrected our text using “p.”. |
|
17. Section 3.3: It seems odd to include multivariate analyses of other COPE factors if they were not considered variables of interest (line 337). Please clarify the rationale. |
Our statistical analyses followed a stepwise approach to guide the identification of variables for inclusion in the final network analysis. We chose this data-driven strategy to ensure that only variables showing meaningful variability (i.e., explained variance) were retained in the final model. Specifically, we included all COPE factors in the initial steps to determine which ones shared variance with demographic/clinical characteristics and self-based processes of change, allowing for a more focused and interpretable final network. |
|
18. Figure 1: Could you include a legend within the figure to indicate what the different labels refer to, so readers do not need to refer back to the text? |
A textual explanation accompanying the visual network analysis was added. See page 14 “Figure 1. The resultant network analysis showing the lasso regularized partial coefficients between SACS items with G-PIPS total scores and COPE active coping scores” |
|
19. Supplementary Figures (S3 and S4): These are difficult to interpret. In S3, the legend does not indicate which nodes are being referred to, and in Figure on page 19, the numbers are not fully visible. Please improve figure clarity and include legends. |
Figure S2 and S3 report the findings from the Network Edge Stability with Bootstrapped confidence Intervals of all Edges findings. To improve the interpretation of the network stability, we added the following text in the 3.4.2. Network Accuracy and stability section , see page 13, lines 415-418. “For the edge stability plot (SM2), for most of the edges (connections between variables) we observed shorter lines (CIs) and CIs not including zero. This suggests that each shorted line edge (not containing zero, too) is consistently present across the 5000 bootstrap samples”.
We also improved the Figure’s 1 visibility. |
|
20. Centrality measures table: Are these values standardised and sorted (e.g., from least to most expected influence)? To further improve clarity, could you include a line or indicator connecting the different data points? |
There is no table named “centrality measures table”. We suspect that the review refers to the Supplementary Material S3: Network’s Centrality Stability Coefficient Intervals. This graph presents the Central Stability Coefficient (CS-coefficient) which indicates the maximum proportion of the sample that can be dropped while still retaining a correlation of at least 0.7 with the original centrality index. The plot displays boxplots for each centrality index (e.g., strength, closeness, betweenness). Each boxplot shows the distribution of correlations between the original centrality values and the values obtained from case-dropping bootstrap samples- not from least to most expected influence (e.g., 10%, 20%, ..., up to 75% of cases dropped). Higher median correlations (boxes near 1.0) indicate better stability with the overall stable centrality matrix to be considered when the correlation between the original centrality values and the values obtained from case-dropping bootstrap samples is retained at least 0.7 as cases are dropped. As we cannot intervene in the actual graph because JASP automatically generate the CS-coefficient graphs, we revised the content in the 3.4.2 section to improve the interpretation of JASP’s generated outcomes, see page 13, lines 418-429 and below “The correlation stability (CS) coefficients (see SM3) for most of the edges were above the minimally acceptable index stability (CS >0.25), suggesting high index stability. However, when we visually inspected the boxplots for the strength’s stability, we observed some steady drop distribution of correlations between the original centrality values and the values obtained from case-dropping bootstrap samples (the correlation should be retained at least 0.7 as cases are dropped). Specifically, the spearman rho median correlation (e.g., the y-axis representing between original and bootstrapped centrality values correlations) was 0.41 for active coping, 0.62 for the psychological inflexibility (G-PIPS-II), and for the key SACS items 2, 9, and 10 were 1.10, 1.20, and 0.71, respectively. These findings indicate some instability in the centrality matrix, proposing a cautious interpretation of the resultant network models”. |
|
21. In the discussion, it is stated that both active coping and three out of six transcending items are identified as the most central nodes (line 446), whereas in the results, active coping is described as the “most central node,” followed by four transcending items. Please ensure consistency between results and discussion. |
Thanks for this. We have now amended the results section as follow, see page 15, lines 399-402. “For the resultant model including the LASSO regularised partial correlation network, which accounted for the inter-relationships between all nodes, indicated that active coping was strongly and positively linked with the centering item #2 (edge value = 0.352) and with the transcending item #9 (edge value = 0.073)”. |
|
22. It would be beneficial to include further discussion on the association between psychological inflexibility and transcending items #9 and #10 (line 451) |
We have now included a new paragraph discussing the theory-based associations between psychological inflexibility and the transcending items #9 and #10. See on page 16, lines 521-541 and below “Likewise, the observed negative association between psychological inflexibility and items from the Transcending sub-scale (#9 and #10) may reflect theoretically grounded mechanisms, involving self-based processes of change. Specifically, this finding may indicate the contribution of healthy “selfing” practices, including self-as-process (i.e., an ongoing discrimination of our behaviours as they occur in the moment-x) and self-as-context (i.e., adopting a transcendent perspective of the self as a stable observer of internal experiences-x) in the management of cancer-related pain. From this perspective, cultivating self-based processes of change may buffer against psychologically inflexible responses by enabling individuals with cancer-related pain to recognize (verbally discriminate-x) and label their ongoing experiences more flexibly (e.g., “right now I am experiencing cancer-related pain that feels debilitating”). This perspective allows for a noticing stance, in which one acknowledges the experience (e.g., “this pain is one of many experiences I have”) without becoming rigidly attached to it. Consistent engagement with self-as-context and the ongoing discrimination of pain -related experiences from this perspective, may offer individuals a sense of a stable psychological vantage point. That is, despite the experiences differing across time, place, and situations, the person who notices these experiences remains the same. This capacity to observe, rather than be overwhelmed by painful content, may be particularly relevant in the context of cancer-related pain, where such experiences are frequently perceived as threatening to the “threads of the self”[28]. There is a need for more experiential and analogous proof-of-concept studies to support this”.
|

Round 2
Reviewer 2 Report
Comments and Suggestions for Authors
Thank you for addressing my previous comments. The manuscript is improved and clearer in several areas.
- References still need further attention. Please ensure that the concepts mentioned refer to the original sources rather than secondary reviews. For example on page 3, line 9”", in the context of chronic pain, there are various conceptualizations and terms related to the self, including self-concept [29], mental defeat [29], self-pain [29], and self-compassion [29].” These should be attributed to the original sources where possible.
- Please check references 25 and 32 for typo’s.
- Point 8: Thank you for providing further context. Please include the expected direction of the correlation in the text.
- Point 15: Thank you for clarifying. Since this is an ordinal variable, it should not be reported as a mean. Instead, it might be more appropriate to rephrase it along the lines of: “The majority of participants (xx%) fell into the 46–55 age range,” or simply refer to the participant descriptive table.
Author Response
Assistant Prof. Vasiliki Eftimiou,
Special Editor-in-chief HealthCare
01/08/2025
Dear Ass. Prof. Eftathiou,
We would like to thank you for inviting us to revise and resubmit our work. We have addressed all of your comments from the second round of this review, and we do hope that this revision will allow you to reach a final decision. We would also like to thank the reviewers for their time spent reviewing this manuscript and their comments, which have greatly improved our manuscript.
Please find below a detailed response to all the reviewers’ comments. We present any changes in the manuscript in a red font.
|
Reviewer 2 |
|
|
1. Please ensure that the concepts mentioned refer to the original sources rather than secondary reviews. For example on page 3, line 9”", in the context of chronic pain, there are various conceptualizations and terms related to the self, including self-concept [29], mental defeat [29], self-pain [29], and self-compassion [29].” These should be attributed to the original sources where possible. |
Thank you for pointing this out. We added the original sources “including self-concept [32], mental defeat [33], self-pain [34], and self-compassion [35].” (page 3 – lines 97,98). |
|
2. Please check references 25 and 32 for typo’s. |
Thank you for pointing this out. We have corrected our references. |
|
3. Point 8: Thank you for providing further context. Please include the expected direction of the correlation in the text. |
We amended this as following, see page 4, lines 162-163: (a) SAC would exhibit significant positive correlations with conventional coping strategies and negative correlations with psychological inflexibility; |
|
4. Point 15: Thank you for clarifying. Since this is an ordinal variable, it should not be reported as a mean. Instead, it might be more appropriate to rephrase it along the lines of: “The majority of participants (xx%) fell into the 46–55 age range,” or simply refer to the participant descriptive table. |
We agree and we have accordingly added this text (page 7, line 319-320): “The present sample included n=135 individuals with cancer-related pain with the majority of participants (43,7%) falling into the 46 to 55 age range.” |
